# ATTENTION PROMPT TUNING

## ABSTRACT

In this paper, we introduce Attention Prompt Tuning (APT) – a computationally efficient variant of prompt tuning for video-based applications. Prompt tuning approaches involve injecting a set of learnable prompts along with data tokens during fine-tuning while keeping the backbone frozen. This approach greatly reduces the number of learnable parameters compared to full tuning. For image-based downstream tasks, normally a couple of learnable prompts achieve results close to those of full tuning. However, videos, which contain more complex spatiotemporal information, require hundreds of tunable prompts to achieve reasonably good results. This reduces the parameter efficiency observed in images and significantly increases latency and the number of floating-point operations (FLOPs) during inference. To tackle these issues, we directly inject the prompts into the keys and values of the non-local attention mechanism within the transformer block. Additionally, we introduce a novel prompt reparameterization technique to make APT more robust against hyperparameter selection. The proposed APT approach greatly reduces the number of FLOPs and latency while achieving a significant performance boost over the existing parameter-efficient tuning methods on the UCF101, HMDB51, and SSv2 datasets for action recognition. *The code and pre-trained models will be publically available after the review process.*

## 1 INTRODUCTION

Transformers Vaswani et al. (2017); Dosovitskiy et al. (2021); Liu et al. (2021) have revolutionized a wide range of fields, including natural language, audio, image, and video processing Han et al. (2022). They have been applied to various downstream tasks, such as language translation Vaswani et al. (2017), audio classification Gong et al. (2022), image classification Dosovitskiy et al. (2021), and action recognition Tong et al. (2022); Bandara et al. (2022). The *de facto* standard when utilizing transformer backbones for downstream tasks is full fine-tuning, i.e. tuning the backbone and a task-specific head by initializing the backbone with pre-trained weights Tong et al. (2022); Bandara et al. (2022); Feichtenhofer et al. (2022). Although full fine-tuning has demonstrated strong performance in most cases, it presents certain disadvantages. These include: (1) *parameter inefficiency*—full fine-tuning can lead to a significant increase in the number of parameters, especially when multiple tasks are involved, as it does not share parameters between tasks; (2) *large labeled data requirements*—full fine-tuning necessitates a substantial volume of labeled data to avoid overfitting and ensure good generalization; (3) *computational demands*—full fine-tuning is computationally intensive and time-consuming, requiring significant computational resources; (4) *overfitting*—there is a risk of overfitting to the new data, particularly if the training data is limited; and (5) *catastrophic forgetting*—fine-tuning on a new task may cause the model to forget previously learned knowledge from the pre-training task.

As a result, alternative methods, including linear probing, adapter tuning Chen et al. (2022), and visual prompt tuning (VPT) Jia et al. (2022), have been proposed, each with its own set of trade-offs (see Fig. 1). *Linear probing* involves tuning only a task-specific head while keeping the backbone parameters fixed (frozen). It is highly parameter-efficient and computationally less demanding. However, its performance is usually limited by the fixed nature of the backbone. On the other hand, *adapter tuning* Chen et al. (2022) introduces learnable Multi-Layer Perceptron (MLP) layers in each transformer block, providing a balance between parameter efficiency and model flexibility. It allows for task-specific adaptations without changing the entire backbone and has shown better performance than linear probing. The added adapter layers increase the number of trainable parameters (but far less

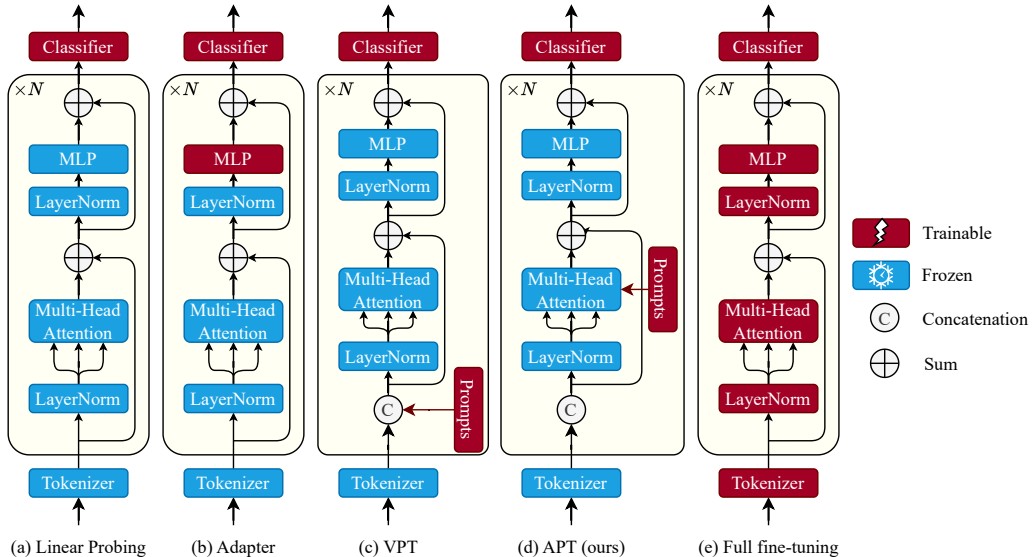

Figure 1: Comparison of our Attention Prompt Tuning (APT) for videos with other existing tuning methods: linear probing, adapter tuning Chen et al. (2022), visual prompt tuning (VPT) Jia et al. (2022), and full fine-tuning.

than full fine-tuning) and FLOPS (hence latency) compared to linear probing. In contrast, *VPT* Jia et al. (2022) adopts tunable prompts that are fed into the model along with the input tokens. These prompts act as task-specific guides, allowing the model to retain the pre-trained knowledge of the backbone while adapting to the new task. VPT is highly flexible, parameter efficient, and has been shown to achieve better performance than linear probing and prompt tuning across a wide variety of downstream tasks specially those involving images Jia et al. (2022). However, we have observed that direct adaptation of VPT for downstream tasks involving videos, such as action recognition, leads to high computation requirements, unstable training, and increased FLOPs during training and inference.

Videos inherently consist of a significantly larger number of tokens compared to images (typically 1568 *vs.* 196 tokens) Tong et al. (2022); Bandara et al. (2022); Qian et al. (2021). Consequently, VPT for videos often necessitates appending a substantial number of prompts (approximately 800 prompts) to the input tokens to achieve performance levels comparable to full fine-tuning. However, this approach presents challenges. The large number of appended prompts increases computational complexity (FLOPs) and latency during both training and inference. Elevated latency is particularly concerning for video-based downstream applications, which already demand significant time and computational resources. As such, the increased latency and FLOPs compromise the flexibility and parameter efficiency of visual prompt tuning for videos, highlighting the importance of reducing computational complexity while maintaining parameter efficiency.

In our experiments we have observed that interactions between video token representations and prompts primarily occur during the multi-head attention (MHA) in the transformer block. This is because both the MLPs and Layer Normalization (LayerNorm) Ba et al. (2016) operations before and after the attention mechanism are applied along the embedding dimension separately for each token/prompt. As a result, the layers preceding the MHA serve as fixed reparameterization for the prompts, rendering the computations on prompts following the attention mechanism redundant. This redundancy arises because the current prompts are discarded at the end of each transformer layer, and new prompts are subsequently appended for the following layer. Motivated by this observation, we propose introducing prompts directly into the attention mechanism, thereby reducing redundancy and computational complexity. Our approach, named Attention Prompt Tuning (APT), minimizes extraneous computations, reduces latency, and enhances the effectiveness of visual prompt tuning for video-based tasks.

Moreover, we noted that VPT is highly sensitive to hyperparameter selection, such as learning rate and weight decay Krogh & Hertz (1991), and often requires a longer fine-tuning time (approximately twice as long) compared to full fine-tuning. This sensitivity is observed in other fields as well, including natural language processing and image processing Liu et al. (2022); Li & Liang (2021). Since video fine-tuning generally takes longer than image fine-tuning, accelerating convergence and enhancing robustness to hyperparameter selection are of great interest. Previous works on prompt tuning Liu et al. (2022) suggest prompt reparameterization as a solution to these challenges, wherein prompts are processed through a reparameterization network consisting of a small MLP layer. However, adding an MLP layer significantly increases the number of parameters and introduces additional computational cost. To address this, we introduce scaling-based reparameterization as a solution that adds a significantly lower number of parameters while incurring negligible computational cost. In this paper, we explore the benefits and effectiveness of this approach for video-based tasks.

## 2 RELATED WORK

**Transformers.** The Transformer architecture Vaswani et al. (2017); Dosovitskiy et al. (2021), originally introduced for natural language processing (NLP) tasks, has been successfully applied to computer vision with the development of the Vision Transformer (ViT) Dosovitskiy et al. (2021). ViT converts images into small patches (tokens) and processes them through multi-head attention modules, enabling tasks such as classification Dosovitskiy et al. (2021); Bandara et al. (2022); Bhojanapalli et al. (2021), segmentation Strudel et al. (2021); Chen et al. (2021); Bandara & Patel (2022), detection Bandara & Patel (2022), synthesis Zhang et al. (2022); Parmar et al. (2018); Chen et al. (2020), and more. With the development of self-supervised pre-training methods such as masked autoencoders for transformers (MAEs) He et al. (2021); Tong et al. (2022); Bandara et al. (2022); Girdhar et al. (2022); Sameni et al. (2022), the use cases of transformers have expanded by leveraging unlabeled data for pre-training, which provides excellent transfer learning capability for various downstream applications with limited supervised labels He et al. (2021); Tong et al. (2022); Bandara et al. (2022). Despite the success of pre-trained transformer models, fine-tuning the entire backbone for downstream tasks is commonly practiced but often inefficient in terms of the number of tunable parameters and computational resources. Consequently, there has been significant interest in exploring parameter-efficient techniques for adopting pre-trained models in downstream applications Jia et al. (2022); Chen et al. (2022); Liu et al. (2022); Li & Liang (2021).

**Efficient Downstream Fine-tuning.** Efficient tuning methods can be divided into two categories: (1) network-based methods and (2) prompting-based methods. Network-based methods, such as AdaptFormer Chen et al. (2022); Pan et al. (2022b)ST-Adapter Pan et al. (2022a), and AIM Yang et al. (2023), introduce a small, learnable network before or after each multi-head attention module, which are fine-tuned while keeping the rest of the parameters fixed. This approach significantly improves parameter efficiency compared to end-to-end tuning. However, the addition of these networks slightly alters the backbone architecture, which is a drawback. In contrast, prompting-based methods Jia et al. (2022), originally inspired by prompt tuning in NLP Liu et al. (2022); Li & Liang (2021), involve inserting a set of learnable prompts along with the input tokens during tuning. Prompt tuning is relatively simple and easily applicable to downstream tasks. However, it has been observed that prompt tuning achieves significantly lower performance than full-tuning in complex video-based applications like action classification. Moreover, prompt tuning requires a higher number of tunable prompts for video applications, resulting in increased latency and computational requirements. In this paper, we eliminate redundant computations of prompt tuning by directly injecting prompts into the attention mechanism.

## 3 METHOD

### 3.1 VISUAL PROMPT TUNING (VPT)

As depicted in Fig. 1, VPT Jia et al. (2022) appends a set of learnable prompts to the input tokens. The combined input (tokens + prompts) is then processed by the Transformer Block, which consists of a sequence of components: LayerNorm, Multilayer Perceptrons (MLPs), and Multi-Head Attention (MHA). Notably, LayerNorm and MLPs operate along the embedding dimension, meaning that

the interaction between the prompts and the input token features occurs primarily within the MHA component. Consequently, the LayerNorm and MLP components preceding the attention mechanism can be viewed as a fixed reparameterization network for the prompts. However, the LayerNorm and MLP components following the MHA are redundant because the current prompts are discarded at the end of the current Transformer Block, and a new set of prompts is appended to the next block. Motivated by this observation, we introduce an optimization to reduce the extra computational redundancy introduced by VPT. Specifically, we introduce the prompts directly into the attention mechanism, thereby saving significant computation – a benefit that is particularly valuable for video-based downstream tasks.

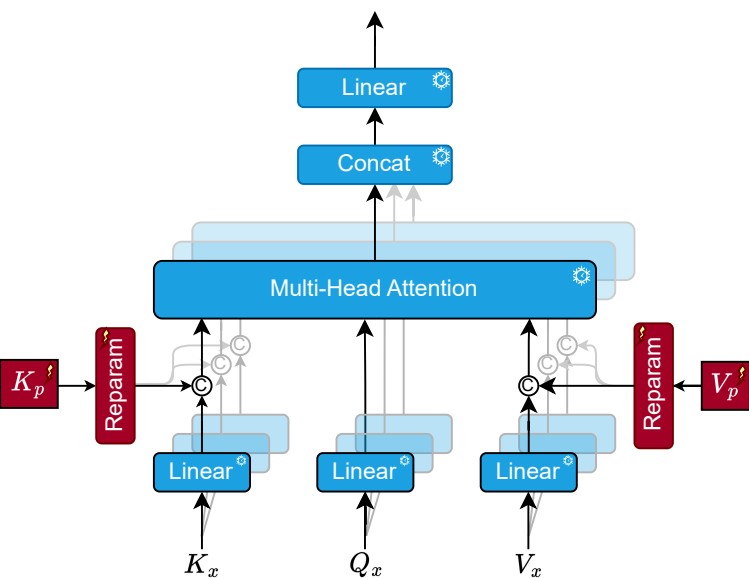

Figure 2: The proposed APT architecture. Modules in **cyan** color are frozen and not updated during fine-tuning. Modules in **purple** color are learned and updated during fine-tuning. Learnable prompts ($K_p$ and $V_p$) are directly injected into the MHA by concatenating them with the Keys ($K_x$) and Values ($V_x$), while the Queries ($Q_x$) remain unchanged.

### 3.2 ATTENTION PROMPT TUNING (APT)

Attention Prompt Tuning (APT) injects learnable prompts directly into the MHA unlike VPT, as shown in Fig. 2. Specifically, we concatenate learnable prompts only with Keys ($K$) and Values ($V$), ensuring that the output from the MHA has the same number of tokens as the input. This avoids the redundant computations by MLP and LayerNorm layers on prompt tokens following the MHA, unlike in VPT.

Formally, the non-local attention operation in MHA is defined as,

$$x = \text{Softmax}\left(\frac{QK^T}{\sqrt{d}}\right)V, \tag{1}$$

where $Q$, $K$, and $V$ denote the Queries, Keys, and Values, respectively, and $d$ is the size of the embedding dimension. In APT, we modify the Keys and Values of the input by appending a set of learnable prompts, denoted by $K_p$ and $V_p$, as follows,

$$K = K_x \oplus K_p, \text{ and } V = V_x \oplus V_p, \tag{2}$$

where $\oplus$ represents concatenation. Here, $K_x$ and $V_x$ are the original Keys and Values derived from the input tokens. The concatenation results in Keys and Values with dimensions of size $(n_x + n_p, d)$, where $n_x$ is the number of input tokens and $n_p$ is the number of prompts. The dimension of Queries ($Q$) remains unchanged at $(n_x, d)$. Consequently, the output after the MHA operation retains the same size as the input, which is $(n_x, d)$.

The efficient design of APT improves computational efficiency by reducing the overhead introduced by VPT, making it particularly advantageous when dealing with large numbers of prompts, as in the case in video-based applications.

## 3.3 PROMPT REPARAMETERIZATION

During our experiments, we observed that the performance of prompt tuning methods is highly sensitive to the proper selection of hyperparameter values, especially learning rate and weight decay. This sensitivity has also been observed in prompt tuning applied to other applications in NLP and image classification. Furthermore, we noticed that prompt tuning typically requires a slightly higher number of fine-tuning epochs compared to full-model fine-tuning. A common practice to make prompt tuning more robust to variations in learning rate and other hyperparameters is to use a reparameterization network, which often involves processing the prompts through an additional MLP network. However, adding an MLP layer to a large number of prompts increases computational complexity, latency, and the number of tunable parameters. Therefore, we introduce a scaled reparameterization method in which each prompt in $K_p$ and $V_p$ is scaled by a learnable scalar. More precisely, the reparameterization of the Key and Value prompts is as follows,

$$\tilde{K}_p = \max(s_k, 1.0) \times K_p \text{ and } \tilde{V}_p = \max(s_v, 1.0) \times V_p, \tag{3}$$

where $s_k$ and $s_v$ are the learnable scalars of size $\mathbb{R}^{n_p}$. Importantly, $s_k$ and $s_v$ are initialized to 1.0. The proposed scaled reparameterization enables the direct adjustment of prompt values through a tunable scaling factor, leading to faster convergence without sacrificing performance. The number of GLOPS are calculated using fvcore flop count tool[1].

## 4 EXPERIMENTAL SETUP

For all of our experiments, we utilize vanilla transformer architectures, specifically ViT-Small and ViT-Base Dosovitskiy et al. (2021); Tong et al. (2022); Bandara et al. (2022). We initialize the model using pretrained weights from the VideoMAE method Tong et al. (2022); Bandara et al. (2022). Unless otherwise noted, we employ the AdamW optimizer Loshchilov & Hutter (2019). The models are fine-tuned for 100 epochs, which includes a 10-epoch warm-up period, followed by a cosine scheduler to reduce the learning rate to zero. As part of our data augmentation strategy, we apply Random Augment Cubuk et al. (2019). The base learning rate is specified for a batch size of 256, and it is automatically scaled for different batch sizes as needed. To expedite fine-tuning, we use a batch augmentation factor of 2, meaning that for each loaded video, we create two random views. For evaluation, we adhere to the common practice of multi-view testing. We use K temporal clips to span the video length (with K=3 for SSv2, 10 for HMDB51, and 5 for UCF101 Tong et al. (2022); Bandara et al. (2022); Sameni et al. (2022)), and for each clip, we take three spatial views to cover the entire image. The final prediction is obtained by averaging the results across all views.

## 5 RESULTS

### 5.1 ABLATION EXPERIMENTS

For all ablation experiments, we employ the ViT-Small (21.947M params.) Dosovitskiy et al. (2021) model pretrained on the SSv2 (Something-Something v2) dataset Goyal et al. (2017). The pretrained weights are obtained from VideoMAE Tong et al. (2022). We report both top-1 and top-5 accuracy metrics for action classification on the test set of the SSv2 dataset Goyal et al. (2017), unless specified otherwise. The best (i.e., default) configuration is highlighted in grey color and used in the main analysis with ViT-Base architecture. During evaluation, we adhere to a common protocol that involves a 2-view, 3-temporal-clip evaluation. This protocol involves extracting multiple temporal clips and views from each video and aggregating the predictions to produce the final classification result.

---

[1]https://github.com/facebookresearch/fvcore/blob/main/fvcore/nn/flop_count.py

**Effect of Prompt length.** Table 1 shows the impact of varying the number of prompts (i.e., prompt length $n_p$) in the proposed APT on the tunable parameters, GFLOPs, and Top-1/Top-5 accuracy on SSv2. Increasing the prompt length initially leads to an improvement in both Top-1 and Top-5 accuracy. The peak Top-1 accuracy of 57.57% and Top-5 accuracy of 84.15% are achieved with 1400 prompts. However, further increasing the number of prompts to 1600 or 2000 does not lead to a significant improvement in accuracy, and in some cases, the Top-1 accuracy slightly decreases. This suggests that there may be an optimal number of prompts for this model and dataset, beyond which the additional prompts do not contribute significantly to performance improvement.

Table 1: Effect of prompt length $(n_p)$ on APT with ViT-Small backbone on SSv2.

| $n_p$ | Params. (M) | Top-1 (%) | Top-5 (%) |
|---|---|---|---|
| 400 | 0.692 (3.2%) | 55.81 | 83.17 |
| 600 | 1.003 (4.5%) | 56.52 | 83.76 |
| 800 | 1.316 (6.0%) | 56.74 | 83.87 |
| 1000 | 1.628 (7.4%) | 57.18 | 84.21 |
| 1200 | 1.940 (8.8%) | 57.44 | 84.24 |
| 1400 | 2.252 (10.3%) | **57.57** | **84.15** |
| 1600 | 2.564 (11.7%) | 57.07 | 84.14 |
| 2000 | 3.188 (14.5%) | 57.55 | 84.03 |

**Effect of Prompt Reparameterization.** As previously discussed, prompt tuning methods are highly sensitive to hyperparameter selection, especially with regards to the choice of learning rate and weight decay parameters. To enhance the robustness of APT to hyperparameter selection, we introduce a prompt reparameterization technique that requires minimal computational overhead and introduces only a small number of tunable parameters. In this analysis, we examine the convergence of training loss during fine-tuning both with and without the application of prompt reparameterization. As illustrated in Fig. 3, utilizing scaled reparameterization leads to faster convergence and a lower training loss compared to the absence of reparameterization. This approach results in improved performance, as evidenced by a **1.02%** gain in top-1 accuracy (**55.81%** vs. **54.85%**) and a **0.93%** gain in top-5 accuracy (**83.17%** vs. **82.24%**). Notably, the observed improvement is achieved with a negligible increase in tunable parameters ($2 \times 400 \times 12 = 9600$, representing only **1.4%** of the total tunable parameters). This enhancement in both performance and convergence speed is particularly valuable for video-based downstream applications, where fine-tuning on video datasets is both time-consuming and computationally demanding.

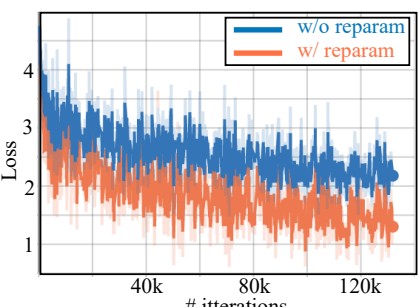

Figure 3: Training loss comparison: with and without prompt reparameterization. The use of scaled reparameterization results in faster convergence and a lower training loss.

**Effect of Dropout on Attention Prompts.** Through experimentation, we found that applying dropout to attention prompts $K_p$ and $V_p$ improved both top-1 and top-5 accuracy. Dropout serves as a regularization technique, reducing overfitting and promoting diverse prompt representations. Table 2 shows performance for different dropout rates, with the best results achieved at a **10% rate**, which balanced informativeness and variability in the prompts. Overall, dropout in prompt tuning enhances classification performance and generalization.

Table 2: Performance for different dropout rates applied to $K_p$ and $V_p$.

| Dropout | Top-1 | Top-5 |
|---|---|---|
| 0% | 55.21 | 82.66 |
| 10% | **57.57** | **84.15** |
| 20% | 52.14 | 80.80 |

**Effect of Random Augmentations.** During full-fine tuning, extensive data augmentations are typically applied through Random Augment Cubuk et al. (2019) to enhance the model's generalization ability by introducing diversity into the training data. However, our experiments with APT, as well as with linear probing and VPT, revealed a different behavior. We observed

Table 3: Effect of Random Augmentations (RA).

| Configuration | top-1 | Top-5 |
|---|---|---|
| No Aug. | 55.81 | 83.17 |
| RA (m2-n2-mstd0.2-inc1) | **55.82** | **83.42** |
| RA(m7-n4-mstd0.5-inc1) | 53.94 | 82.23 |

that applying the same extent of Random Augmentations during APT, linear probing, and VPT did not lead to performance improvements and, in some instances, resulted in diminished performance. We hypothesize that this behavior is attributable to the fact that the backbone remains frozen during APT, linear probing, and VPT, thereby limiting the capacity for overfitting. As training relies on a relatively small fraction of tunable parameters, the need for aggressive data augmentation is reduced.

**Effect of Weight Decay (WD).** During the fine-tuning of APT, we investigated the impact of weight decay regularization on model performance. Our experiments revealed that the choice of weight decay value also plays a crucial role in the effectiveness of APT.

Specifically, we observed that applying an excessively high weight decay value (like $1e-2$ like utilized in full-tuning) led to a deterioration in performance, likely due to over-constraining the attention prompts values and which hinders its ability to change the input token representations in the desired direction. Conversely, completely removing weight decay regularization (i.e., setting weight decay to zero) also resulted in sub-optimal performance, as the attention prompts become more susceptible to overfitting to the training data.

Table 4: Effect of WD.

| WD | Top-1 | Top-5 |
|---|---|---|
| 1e-2 | 51.43 | 79.98 |
| 1e-4 | 55.47 | **83.35** |
| 1e-5 | **55.81** | 83.17 |
| 0 | 55.24 | 83.05 |

Through hyperparameter tuning, we determined that a moderate weight decay value of $5e-3$ yielded improved performance, achieving an optimal balance between preventing overfitting and allowing flexibility to attention prompts. The performance results for different weight decay values are summarized in Table 4.

**Effect of Attention Prompt Placement.** In the default configuration of APT, we append attention prompts to the MHA at each Transformer Block within the ViT. While this placement strategy yields the best performance, our experiments reveal that the placement of attention prompts at different depths of the Transformer Blocks has varying contributions to the overall performance. Specifically, we find that attention prompts placed at deeper Transformer Blocks (i.e., blocks closer to the output layer) have a more substantial impact on performance compared to prompts placed at shallower Transformer Blocks (i.e., blocks closer to the input layer). This observation suggests that refining feature representations at deeper transformer layers is more important for downstream performance than the features at shallow transformer layers. Figure 4 illustrates the performance improvement achieved by incrementally appending attention prompts, starting from the deepest Transformer Blocks and progressively extending the placement to shallower blocks. The results highlight the importance of attention prompt placement in the APT framework and its influence on model performance.

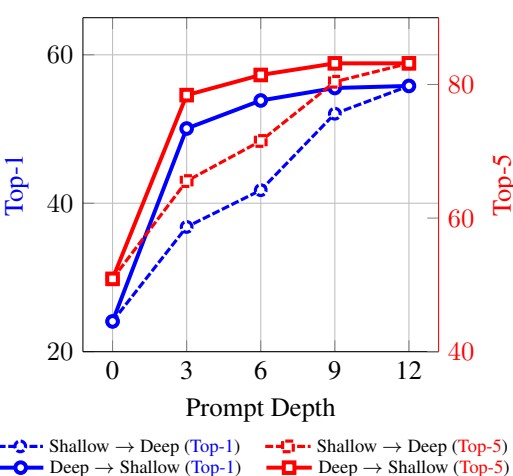

Figure 4: Effect of prompt depth with APT.

## 5.2 COMPUTATIONAL COMPLEXITY

We compare computational and parameter efficiency of APT with VPT Jia et al. (2022) in Fig. 5. The results demonstrate that injecting prompts directly into the multi-head attention significantly reduces the number of tunable parameters and GLOPs, unlike VPT, which concatenates prompts with the input tokens. The reduced number of GFOPs of APT is beneficial for more efficient and scalable deployment of pre-trained transformer models for video applications.

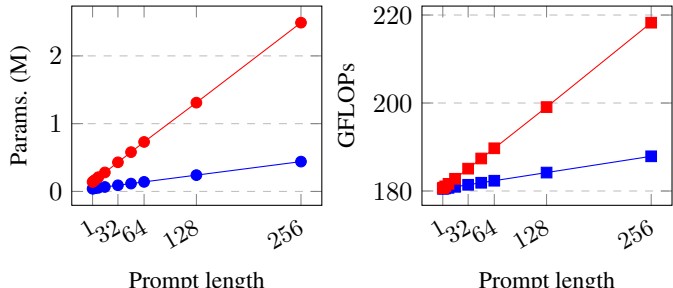

Figure 5: Comparison of tunable params. (left) and GFLOPs (right) with prompt length for **VPT** and **APT (Ours)**.

Table 5: Comparison of APT performance with other parameter efficient tuning methods. We use VideoMAE Tong et al. (2022); Bandara et al. (2022) pre-trained ViT-Base backbone as the initialization.

| Method | Tuned Params. (M) | GFLOPs | SSv2 | | UCF101 | | HMDB51 | |
|---|---|---|---|---|---|---|---|---|
| | | | top-1 | top-5 | top-1 | top-5 | top-1 | top-5 |
| Linear-Probing | 0.07 (0.08%) | 180.03 | 29.23 | N/A | 94.3 | 99.5 | 49.84 | 75.9 |
| VPT-1 | 0.08 (0.09%) | 180.50 | 43.73 | N/A | 94.1 | N/A | 52.76 | N/A |
| VPT-5 | 0.12 (0.14%) | 180.60 | 45.87 | N/A | 94.7 | N/A | 53.66 | N/A |
| AdaptFomrer-1 | 0.10 (0.12%) | 180.51 | 50.03 | N/A | 94.3 | 99.5 | 51.68 | 78.3 |
| AdaptFomrer-4 | 0.15 (0.17%) | 180.60 | 54.70 | N/A | 94.5 | 99.5 | 51.81 | 78.9 |
| AdaptFomrer-64 | 1.26 (1.46%) | 182.33 | 59.02 | N/A | 94.5 | 99.7 | 55.69 | 79.4 |
| APT-1 (ours) | 0.081 (0.09%) | 180.51 | 42.92 | 72.18 | 96.4 | 99.8 | 58.23 | 83.09 |
| APT-5 (ours) | 0.087 (0.09%) | 180.63 | 49.15 | 78.50 | 96.5 | 99.9 | 61.43 | 87.12 |
| APT-100 (ours) | 0.235 (0.27%) | 183.38 | 57.15 | 84.85 | 97.6 | 100.0 | 67.58 | 89.67 |
| APT-200 (ours) | 0.391 (0.45%) | 186.27 | 59.43 | 86.39 | **97.7** | **100.0** | 68.17 | 88.63 |
| APT-400 (ours) | 0.703 (0.81%) | 192.05 | **60.79** | **87.15** | 97.7 | 99.9 | **70.12** | **90.78** |
| Full-tuning | 86.36 (100%) | 180.03 | 70.8 | 92.4 | 96.1 | 99.4 | 73.3 | N/A |

### 5.3 MAIN ANALYSIS

### 5.4 RESULTS WITH VIDEOMAE PRE-TRAINED BACKBONES

Table 5 compares the results of our APT with linear probing, full-tuning, and existing parameter-efficient methods (VPT Jia et al. (2022) and AdaptFormer Chen et al. (2022)) on three action recognition datasets: SSv2 Goyal et al. (2017), UCF101 Soomro et al. (2012), and HMDB51 Kuehne et al. (2011). Note that, for these experiments, we utilize VideoMAE Tong et al. (2022) pre-trained backbones which is currently the state-of-the-art pre-trained models for ViT backbones.

For the UCF101 and HMDB51 datasets, our APT achieves results close to full-tuning while using less than 1% of tunable parameters. Notably, for UCF101, APT achieves higher accuracy than full-tuning (97.7% vs. 96.1% top-1) with only 200 attention prompts, equivalent to 0.45% of tunable parameters. On the HMDB51 dataset, APT results are slightly lower than full-tuning (70.12% vs. 73.3%), but still very close. However, APT outperforms VPT and AdaptFormer by a significant margin, reducing the performance gap between full-tuning and parameter-efficient methods.

When comparing results on the SSv2 dataset, which is larger and has more complex action classes, there is approximately a 10% performance gap between APT and full-tuning. This difference may be attributed to the complexity of the action classes in the dataset. However, when comparing APT

with VPT and AdaptFormer, APT achieves better results with significantly fewer tunable parameters (60.79% top-1 with 0.81% tunable parameters vs. 59.20% top-1 with 1.46% tunable parameters).

Overall, APT significantly reduces the gap between full-tuning and existing parameter-efficient methods, establishing itself as the new state-of-the-art for parameter-efficient tuning in action recognition. Additionally, APT further reduces the number of tunable parameters required to adopt pre-trained transformer backbones for video-based downstream applications.

## 5.5 RESULTS WITH OTHER PRE-TRAINED BACKBONES

| Method | Tune Params. (M) | Pre-training method | |
| --- | --- | --- | --- |
| | | CLIP | Supervised K400 |
| Linear Probing | 0.07 (0.08%) | 21.9 | 29.3 |
| VPT | 0.08 (0.09%) | 45.1 | 33.1 |
| AdaptFormer-4 | 0.15 (0.17%) | 59.1 | 42.3 |
| AdaptFormer-64 | 1.3 (1.46%) | 63.4 | 46.0 |
| ST-Adapter | 7.20 (8.33%) | 66.3 | NA |
| AIM | 14.3 (16.56%) | 66.4 | NA |
| APT-200 (ours) | 0.39 (0.45%) | 64.8 | 47.2 |
| APT-400 (ours) | 0.70 (1.17%) | **66.8** | 50.3 |
| APT-600 (ours) | 1.01 (1.17%) | **66.8** | **50.4** |
| Full-tuning | 86.36 (100%) | 66.9 | 55.7 |

Table 6: The comparison of APT with other parameter-efficient methods using other pre-trained approaches, such as CLIP Radford et al. (2021) and supervised pre-training on K400 Kay et al. (2017) with the ViT-Base backbone on the SSv2 dataset. Reported results indicate top-1 accuracy.

To demonstrate that the proposed APT is not dependent on a specific pre-trained method and performs well with other pre-trained approaches, such as CLIP and supervised pre-trained models, we conducted experiments on SSv2 using the ViT-Base backbone. The results of these experiments are summarized in Table 6. From the data presented in the table, it is evident that the proposed APT achieves SOTA results even when compared to other pre-trained approaches, all while maintaining parameter efficiency.

*Taken together, all the aforementioned results affirm APT's performance as a better parameter-efficient training method than existing approaches, particularly well-suited for handling spatiotemporal data – which often involves training with a large number of tokens than images.*

## 6 CONCLUSION

In this paper, we proposed APT, a variant of VPT, for efficient adaptation of pre-trained transformer models in downstream tasks. While VPT offers flexibility and can be easily adapted to various applications, it requires a significant number of prompts to achieve satisfactory results in video-based tasks like action classification, leading to increased FLOPs and latency during inference and hindering parameter efficiency. In contrast, our APT approach injects attention prompts directly into the attention mechanism, resulting in reduced FLOPs and improved performance compared to VPT and other parameter-efficient methods like AdaptFormer. We also introduced a novel prompt reparameterization technique to enhance tuning convergence and robustness to hyperparameter selection. Overall, APT achieves superior results, sometimes surpassing full-tuning, with high parameter efficiency and minimal increase in GLOPs compared to existing approaches. However, a limitation of APT and other parameter-efficient methods is the potential increase in FLOPs and latency during deployment, which warrants further exploration as future work.

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
