# ATTENTION PROMPT TUNING

## 1 EFFECT OF DROPOUT ON ATTENTION PROMPTS.

Figure 1 illustrates the curves for Top-1 accuracy, Top-5 accuracy, and validation loss at different dropout levels (0%, 10%, and 20%). The figures reveal that, initially, the accuracies for both Top-1 and Top-5 were higher when no dropout was applied. However, as training progressed, the results improved significantly with a dropout rate of 10%. This indicates that utilizing dropout in attention prompts effectively prevents the network from overfitting to the prompts and plays a crucial role in optimizing attention prompt tuning to achieve superior test results.

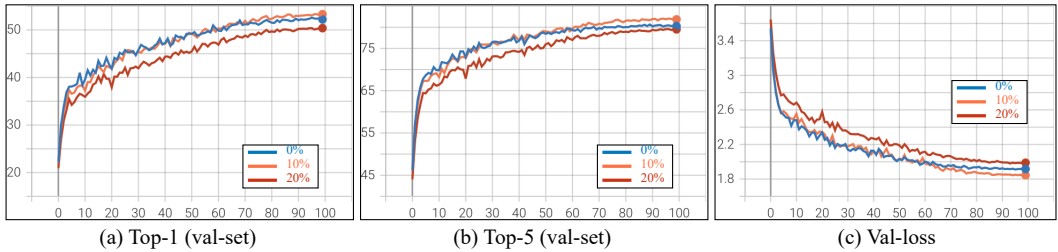

Figure 1: Effect of dropout on attention prompts. Applying 10% dropout to attention prompts results in the best performance on both validation and test sets.

## 2 EFFECT OF RANDOM AUGMENTATIONS.

Figure 2 showcases the top-1 accuracy, top-5 accuracy, and validation loss for different levels of augmentations applied during attention prompt tuning. Previous studies on efficient prompt tuning, such as linear probing and adapter tuning **?**, did not incorporate augmentations. This omission was due to the difficulty of optimization or the inferior results when augmentations were applied. However, in the case of full-tuning, Auto-Augmentation is employed with a magnitude of 7 (i.e., `m7-n4-msd0.5-inc1`). We also observed that applying higher-magnitude augmentations during attention prompt tuning led to poorer performance compared to no augmentations. On the other hand, when utilizing lower-magnitude augmentations (i.e., `m2-n2-msd0.2-inc1`), we achieved the best results.

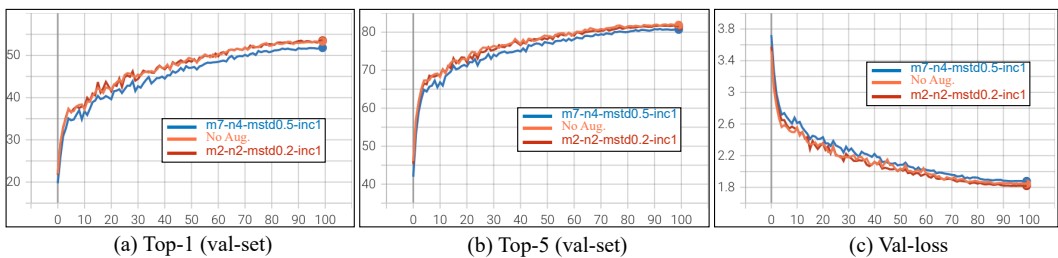

Figure 2: Effect of Random Augmentations.

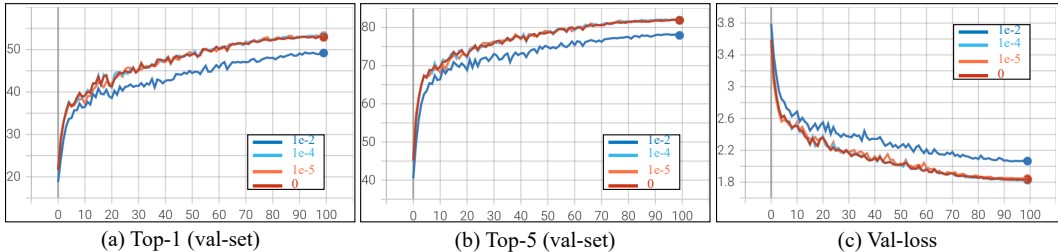

|  | (a) Top-1 (val-set) | (b) Top-5 (val-set) | (c) Val-loss |

Figure 3: Effect of weight decay regulariztion on attention prompts. Applying small weight decay regularization results in better performance.

## 3 EFFECT OF WEIGHT DECAY REGULARIZATION

Figure 3 illustrates the top-1 accuracy, top-5 accuracy, and validation loss curves corresponding to different magnitudes of weight decay regularization. Similar to prompt tuning in the NLP domain, we observed that attention prompt tuning is highly sensitive to weight decay regularization and has a substantial impact on the final results. As depicted in the figure, employing higher weight decay regularization (1e-2) leads to significantly poorer performance compared to a lighter weight decay. In general, we observed that a very small weight decay regularization tends to favor attention prompt tuning. Although the differences in performance are minor, we obtained the best results with a weight decay regularization of 1e-4 compared to no weight decay.

## 4 TRAIN/VAL CONFIGURATIONS FOR DIFFERENT DATASETS

**HMDB51 ?**   Table 1 details the training setting used to produce the results on HMDB51. For more details, please refer the code-base.

**UCF101 ?**   Table 2 details the training setting used to produce the results on UCF101. For more details, please refer the code-base.

**SSv2 ?**   Table 3 details the training setting used to produce the results on SSv2. For more details, please refer the code-base.

## 5 MODEL INITIALIZATION

For all of our experiments, we initialized our model with VideoMAE **??** pre-trained weights on the respective dataset. The links to VideoMAE pre-trained models are available here: `https://github.com/MCG-NJU/VideoMAE/blob/main/MODEL_ZOO.md`.

## 6 RELEASE OF THE CODE

We will make the code and fine-tuned models publically available after the review process.

Table 1: Attention prompt tuning setting for HMDB51 dataset.

| Parameter | Value |
|---|---|
| batch size | 8 |
| batch size val | 8 |
| epochs | 200 |
| model | vit-base-patch16-224 |
| tubelet-size | 2 |
| input-size | 224 |
| optimizer | adam |
| optimizer eps | 1e-08 |
| momentum | 0.9 |
| weight-decay | 1e-05 |
| lr | 0.3 |
| warmup-lr | 1e-06 |
| min-lr | 1e-06 |
| warmup-epochs | 5 |
| AutoAugment | true |
| color-jitter | 0.4 |
| AutoAugment | rand-m7-n4-mstd0.5-inc1 |
| smoothing | 0.1 |
| test-num-segment | 10 |
| test-num-crop | 3 |
| prompt_num_tokens | {1,5,100,200} |
| prompt-start | 0 |
| prompt-end | 11 |
| prompt-dropout | 0.1 |
| prompt-init | random |
| transfer-type | prompt |
| prompt-reparam | true |

Table 2: Attention prompt tuning setting for UCF101 dataset.

| Parameter | Value |
|---|---|
| batch size | 8 |
| batch size val | 8 |
| epochs | 200 |
| model | vit-base-patch16-224 |
| tubelet-size | 2 |
| input-size | 224 |
| optimizer | adam |
| optimizer eps | 1e-08 |
| momentum | 0.9 |
| weight-decay | 1e-05 |
| lr | 0.05 |
| warmup-lr | 1e-06 |
| min-lr | 1e-06 |
| warmup-epochs | 5 |
| AutoAugment | true |
| color-jitter | 0.4 |
| AutoAugment | rand-m4-n2-mstd0.2-inc1 |
| smoothing | 0.1 |
| test-num-segment | 5 |
| test-num-crop | 3 |
| prompt_num_tokens | {1,5,100,200} |
| prompt-start | 0 |
| prompt-end | 11 |
| prompt-dropout | 0.1 |
| prompt-init | random |
| transfer-type | prompt |
| prompt-reparam | true |

Table 3: Attention prompt tuning setting for SSv2 dataset.

| Parameter | Value |
|---|---|
| batch size | 8 |
| batch size val | 8 |
| epochs | 200 |
| model | vit-base-patch16-224 |
| tubelet-size | 2 |
| input-size | 224 |
| optimizer | adam |
| optimizer eps | 1e-08 |
| momentum | 0.9 |
| weight-decay | 1e-05 |
| lr | 0.3 |
| warmup-lr | 1e-06 |
| min-lr | 1e-06 |
| warmup-epochs | 5 |
| AutoAugment | true |
| color-jitter | 0.4 |
| AutoAugment | rand-m2-n2-mstd0.2-inc1 |
| smoothing | 0.1 |
| test-num-segment | 10 |
| test-num-crop | 3 |
| prompt-num-tokens | {1,5,100,200} |
| prompt-start | 0 |
| prompt-end | 11 |
| prompt-dropout | 0.1 |
| prompt-init | random |
| transfer-type | prompt |
| prompt-reparam | true |