# OpenReview forum: "Attention Prompt Tuning"
_ICLR.cc/2024/Conference — ICLR 2024 Conference Withdrawn Submission_

### Official Review · Reviewer_HmAo · 2023-10-21

**Soundness:** 3 good
**Presentation:** 3 good
**Contribution:** 2 fair
**Rating:** 6
**Confidence:** 4

**Summary:**

This paper mainly focuses on prompt tuning for video analysis. The authors propose an attention prompt tuning method to reduce the number of parameters needed in finetuning. Besides, they propose an additional regularization term to enhance the capacity of learnable prompts for better training. Experimental results demonstrate that this new method outperforms Visual Prompt Tuning in video recognition tasks in benchmarks such as Something-Something, UCF-101, and HMDB-51.

**Strengths:**

1. The proposed method is simple and effective.
2. The methodology is clearly explained, the motivation behind the proposed solution is properly justified and the overall method seems intuitive.
3. The proposed method shows favorable results in comparison with previous prompting and adaptation techniques.
4. The paper is easy to read and well written.

**Weaknesses:**

1. Larger-scale experiments are needed to validate the proposed parameter-efficient tuning as an alternative to full fine-tuning. For models like ViT-S and ViT-B, the results on SSV2 are significantly better, and the cost of fine-tuning is still affordable, so the advantage of the proposed method is not clear. (Cheaper training but sacrificing performance)
2. It would be better to compare the proposed method with other VPT variants, such as [1].


[1] Han C, Wang Q, Cui Y, et al. E^ 2VPT: An Effective and Efficient Approach for Visual Prompt Tuning[J]. ICCV 2023.

**Questions:**

Please refer to the weaknesses.

---

### Official Review · Reviewer_fMwu · 2023-10-29

**Soundness:** 2 fair
**Presentation:** 3 good
**Contribution:** 2 fair
**Rating:** 3
**Confidence:** 5

**Summary:**

This paper introduces a computationally efficient prompt tuning method, known as Attention Prompt Tuning (APT), designed for video-based tasks. APT directly adds prompts to the key and value tokens, reducing computational overhead. Furthermore, APT introduces a re-parameterization technique involving the scaling of prompt tokens. Experiments conducted on action recognition demonstrate the effectiveness of APT.

**Strengths:**

1. The problem addressed in the paper is critical, and the motivation behind APT is compelling. Traditional prompt tuning methods, such as VPT, encounter high computational overhead, and applying VPT to video-based tasks exacerbates this burden. Moreover, VPT is truly sensitive to hyperparameters, including but not limited to the number of prompt tokens, learning rate, and so on.

2. The proposed APT is both concise and effective. It appears that VPT can be enhanced to APT with just a few lines of code, making APT user-friendly for video-based tasks.

**Weaknesses:**

1. The paper lacks an in-depth analysis of the proposed method. The authors should conduct a more comprehensive explanation of the
re-parameterization technique they proposed. For example,the authors should clearly explain why re-parameterization can enhance robustness and convergence speed while reducing loss and what is the fundamental distinction between the proposed re-parameterization and the standard re-parameterization method [1].

2, Some of the experimental results are confusing. The authors claim that the proposed re-parameterization technique can improve hyper-parameter robustness. However, there is a lack of supporting experiments. APT still exhibits sensitivity to prompt length (Table 1) and dropout rate (Table 2). The only comparison between with and without re-parameterization is about the loss curve, which may not adequately demonstrate robustness.

3. Regarding generalizability, the paper only conducts experiments on the standard ViT architecture and lacks experiments on more advanced ViT architectures. Furthermore, since APT is both effective and efficient compared to VPT, the authors could also evaluate it across various modalities to demonstrate its generalizability.

[1] Xiao Liu, Kaixuan Ji, Yicheng Fu, Weng Tam, Zhengxiao Du, Zhilin Yang, and Jie Tang. P-Tuning: Prompt Tuning Can Be Comparable to Fine-tuning Across Scales and Tasks. In ACL'22.

**Questions:**

1. I am curious as to why re-parameterizing prompt tokens can enhance convergence speed and robustness. It would be valuable if the authors could provide an explanation for this effect.

2. In addition to the number of learnable parameters, does the proposed scale-based re-parameterization offer any performance or convergence speed advantages when compared to standard re-parameterization?

---

### Official Review · Reviewer_r3ry · 2023-10-31

**Soundness:** 3 good
**Presentation:** 3 good
**Contribution:** 1 poor
**Rating:** 5
**Confidence:** 4

**Summary:**

In this paper, authors investigate strategies for efficient tuning of Video ViTs and propose to append learnable tokens to key and value within self-attention blocks. Additionally, a simple scaled reparameterization technique is applied to stabilize APT toward hyperparameter selection. The efficacy of APT is validated with experiments of transferring pre-trained VideoMAE and CLIP to SSv2, UCF101 and HMDB51.

**Strengths:**

1. The paper is well-written with clear motivations and sufficient ablation studies.

2. APT achieves promising performance on downstream fine-tuning with small amounts of parameters. Besides, the paper involves comprehensive comparisons with other state-of-the-art methods of different settings.

**Weaknesses:**

1. My main concern lies in that from my perspective, APT is highly similar to prefixing tuning[1-2],  which has already been verified effective for visual tasks in previous works [3-4]. Additionally, a similar reparameterization strategy for prompt tuning has already been explored in [5] (authors also DO NOT cite this method in the paper). In light of previous works, novelty and contribution of APT are substantially restricted.

2. To compare with other baselines, it would be appreciated to include views (temporal and spatial ones) in the table for video models.

3. As for efficient tuning on videos, it is important to not only ensure tunable parameters as few as possible but also consider keeping to low GFLOPS, which may also have great impacts on efficiency, training costs and application value. On that basis, APT may improve performance at the cost of high GFLOPS, e.g., APT-400 outperforms other PEFT methods in Table 5 while bringing about the most increase in GLOPS and in Table 6, AIM hits 66.4 with GFLOPS of 624 while APT-400 operates with the GFLOPS of 192.05*3*3 = 1728.45.

4. The experiments involved in the paper are based mainly on ViT-S and ViT-B, which may be due to some practical limitations. Whereas the performance of delta tuning on large models is more valuable to the community.

[1] AIM: Adapting Image Models for Efficient Video Action Recognition, Li et al. ACL 21.

[2] Towards a Unified View of Parameter-Efficient Transfer Learning, He et al. ICLR 22.

[3] Towards a Unified View on Visual Parameter-Efficient Transfer Learning, Yu et al. Arxiv 22.

[4] FedPerfix: Towards Partial Model Personalization of Vision Transformers in Federated Learning, Sun et al. ICCV 23.

[5] MSP: Multi-Stage Prompting for Making Pre-trained Language Models Better Translators, Tan et al. ACL 22.

**Questions:**

Please refer to the Weaknesses part first.

Additionally, according to VPT, VPT-deep performs better than VPT-full and VPT-shallow. What type is the VPT baseline referred to in the paper?  In the meantime, employing APT in deep layers is also more effective than in shallow layers, as displayed in Figure 4. However, how large the performance gap between APT-deep, APT-shallow and APT-full remains unknown.

---

### Official Review · Reviewer_nevf · 2023-11-01

**Soundness:** 1 poor
**Presentation:** 3 good
**Contribution:** 1 poor
**Rating:** 3
**Confidence:** 4

**Summary:**

For video-based applications, normal prompt tuning increases latency and the number of floating-point operations (FLOPs) during inference, reducing parameter efficiency compared to images. To address these challenges, this paper introduces Attention Prompt Tuning (APT), where prompts are directly injected into the keys and values of the non-local attention mechanism within the transformer block. Additionally, a novel prompt reparameterization technique is introduced to enhance APT's robustness against hyperparameter selection. The proposed approach reduces FLOPs while achieving performance improvements over existing parameter-efficient tuning methods for action recognition.

**Strengths:**

1. The writing is clear.
2. The discussed problem is special in video-based applications.

**Weaknesses:**

1. The proposed method lacks innovation. Firstly, the prompt in existing methods can be inserted not only after the tokenizer, so Figure 1 is problematic. Secondly, there have been numerous studies discussing the position of prompts in the vision transformer, with basic choices including Prompt Tuning (Pro-T) [1] and Prefix Tuning (Pre-T) [2]. The Pre-T technique has been widely used in existing computer vision methods, e.g., [3]. Is there any difference between the first technique of this paper and Prefix Tuning?

[1] The Power of Scale for Parameter-Efficient Prompt Tuning, EMNLP 2021.

[2] Prefix-Tuning: Optimizing Continuous Prompts for Generation, ACL 2021.

[3] DualPrompt: Complementary Prompting for Rehearsal-free Continual Learning, ECCV 2022.

2. The authors do not connect the techniques proposed in this paper and video. Why can these proposed techniques solve the special problem of prompt tuning in video applications? The performance of Pre-T is inherently better than that of Pro-T. Is the performance improvement derived from Pre-T itself or from solving the problem?

3. There is no detailed analysis on how much parameter reduction is achieved compared to MLP-based reparameterization, nor any changes in performance.

4. There should be a fine-grained control of variables for each model (GFLOPS and performance under fine-tuning different parametric quantities should be compared in a more granular way).

In conclusion, this paper is more similar to a technical report, rather than a research paper. It applies a widely used trick Prefix Tuning to prompt tuning in video-based applications, and treats it as the core contribution of this paper. More importantly, this paper lacks the analysis for the special problem in video-based applications (i.e., need many prompts) and lacks the insights to design their techniques for solving this special problem. Pre-T has inherently better performance than Pro-T, so the performance improvement comes from Pre-T technology itself or from solving the problem? In addition, regarding the second contribution, why do the authors not provide the comparison between their prompt reparameterization technique and existing prompt reparameterization techniques? I believe that this paper falls considerably short of the standard required for ICLR and, as such, I recommend rejection.

**Questions:**

1. What is the main difference between the first proposed technique and Pre-T?

2. What is the relationship between the proposed techniques and video-based applications? Are there any special properties of video-based applications to be considered for designing the techniques?

3. Why do the authors analyze the "Effect of Prompt Reparameterization"? The focus is on the comparison between the Prompt Reparameterization proposed in this paper and existing Prompt Reparameterization techniques, and the effectiveness of Prompt Reparameterization itself has already been proven.

4. Can the authors provide more detailed analyses on GFLOPS and performance about their method compared to existing method? Figure 5 compares VPT and APT on GFLOPs, but I cannot find the performance comparison. Table 5 provides the comparison of different methods on GFLOPs and performance. However, for VPT, only VPT-5 is reported. Compared to VPT-5, APT-5 does no have the advantage on GFLOPs.